



# Impact of dust deposition on the albedo of Vatnajökull ice cap, Iceland.

**Monika Dragosics** [1], Christine D Groot Zwaaftink [2], Louise Steffensen Schmidt [1], Sverrir Guðmundsson [1,3], Finnur Pálsson [1], Olafur Arnalds [4], Helgi Björnsson [1], Throstur Thorsteinsson [1], Andreas Stohl [2]

[1] Institute of Earth Sciences, University of Iceland, , Reykjavik, Iceland; [2] NILU - Norwegian Institute for Air Research, Kjeller, Norway, [3] Keilir, Institute of Technology, Reykjanesbær, Iceland; [4] Agricultural University of Iceland, Hvanneyri, Iceland

*Correspondence to*: Monika Dragosics (mod3@hi.is)

Abstract

Deposition of small amounts of airborne dust on glaciers causes positive radiative forcing and enhanced melting due to the reduction of surface albedo. To study the effects of dust deposition on the mass balance of Brúarjökull, an outlet glacier of the largest ice cap in Iceland, Vatnajökull, a study of dust deposition events in the year 2012 was carried out. The dust-mobilization module FLEXDUST was used to calculate spatiotemporally resolved dust emissions from Iceland and the dispersion model FLEXPART was used to simulate atmospheric dust dispersion and deposition. We used albedo measurements at two automatic weather stations on Brúarjökull to evaluate the dust impacts. Both stations are situated in the accumulation area of the glacier, but the lower station is close to the equilibrium line. For this site (~1210 m a.s.l.), the dispersion model produced 10 major dust deposition events and a total annual deposition of 16 g m$^{-2}$. At the station located higher on the glacier (~1525 m a.s.l.), the model produced nine dust events, with one single event causing ~5 g m$^{-2}$ dust deposition and a total deposition of ~9 g m$^{-2}$ yr$^{-1}$. The main dust source was found to be the Dyngjusandur floodplain north of Vatnajökull; northerly winds prevailed 80% of the time at the lower station when dust events occurred. In all of the simulated dust events, a corresponding albedo drop was observed at the weather stations. The influence of the dust on the albedo was estimated by using the regional climate model HIRHAM5 to simulate the albedo of a clean glacier surface without dust. By comparing the measured albedo to the modelled albedo, we determine the influence of dust events on the snow albedo and the surface energy balance. We estimate that the dust deposition caused an additional 1.1 m w.e. (water equivalent) of snow melt (or 42% of the 2.8 m w.e. total melt) compared to a hypothetical clean glacier surface at the lower station, and 0.6 m w.e. more melt (or 38% of the 1.6 m w e. melt in total) at the station located further upglacier. Our findings show that dust has a strong influence on the mass balance of glaciers in Iceland.

Key words: dust events, glacier, energy balance, snow melt, surface melt, FLEXPART, albedo



1. Introduction

The cryosphere is an important part of the global climate system. Small changes in reflected and
absorbed radiation at snow or ice surfaces can have large impacts on the state of the cryosphere,
and on Earth's climate and its hydrological cycle (e.g., Budyko, 1969, Flanner et al., 2007, Painter et
al., 2013). Albedo, the reflectivity of a surface, is a dominant component of the surface energy
balance. The albedo of snow depends e.g. on the snow grain size, wetness and impurities in the near-
surface snow layer (e.g. Wiscombe and Warren, 1980; Meinander et al., 2014). Estimation of snow
albedo is important to predict seasonal snowmelt and runoff rates and for calculating the regional
and global energy budget. The snow-albedo feedback, where radiation absorption is enhanced due
to impurity content in snow and ice is indicated by complex processes (Hansen and Nazarenko, 2004;
Myhre et al., 2013). This initiates a positive feedback loop, i.e. more snow melt results in more
absorbed radiation which in turn amplifies the melting. Even though direct global radiative forcing of
mineral dust in the atmosphere is calculated as negative in the IPCC report (IPCC, 2013), regionally
this depends on both the optical properties of the dust, deposited amounts and the albedo of the
underlying surface. Icelandic volcanic dust (mostly from basaltic material) is darker and more
absorbing than mineral dust from most other regions. It is expected to cause positive radiative
forcing, due to its dark colour, the high albedo of snow and ice, and a *clumping mechanism*, where
fine dust impurities in snow form larger particles (Dagsson-Waldhauserova et al., 2015) and
accelerate snow melt. In this study, the term *radiative forcing* means the instantaneous surface
enhanced absorption due to deposited dust (Painter et al., 2007). In its effect on snow albedo, dust is
somewhat similar to black carbon (Yoshida et al., 2016, Goelles et al., 2015) which has received much
interest recently as a short-lived climate forcer, especially in the Arctic (e.g. Quinn et al., 2008;
AMAP, 2015; Meinander et al., 2016). Other studies, e.g. Di Mauro et al. (2015) and Zhao et al.
(2014), have shown the impact of dust and black carbon and their effect on radiative forcing and
energy balance. Painter et al. (2007) has shown that snow cover duration in a mountain range in the
United States was shortened through surface shortwave radiative forcing by deposition of desert
dust. Similarly, Flanner et al. (2014) have shown that the snow albedo effect of deposited volcanic
ash from an eruption in Iceland could counteract the otherwise negative radiative forcing of volcanic
eruptions caused by sulphur emissions.

Sources of dust in Iceland are the proglacial areas and sandy deserts which cover more than 22% of
Iceland (Arnalds et al., 2001). Iceland is one of the most active aeolian places on Earth, even though it
is not situated in an arid climate (Arnalds et al., 2016). Due to the large area of sandur plains and



strong winds resulting in numerous dust events. On average, 135 dust days per year occurred in Iceland, with 101 dust days in south Iceland and 34 dust days in northeast Iceland, where a *dust day* is defined as a day when at least one weather station recorded at least one dust observation. (Dagsson-Waldhauserova et al., 2013). Airborne redistribution of dust has a strong influence on

climate, snowmelt and Icelandic soils. Satellite images have shown that dust particles can be transported over the Atlantic and Arctic Ocean, sometimes for more than 1000 km (Arnalds, 2010). Therefore, Icelandic dust is likely to contribute to Arctic or European air pollution and can affect the climate via dust deposition on Arctic glaciers or sea ice (Arnalds et al., 2016). Icelandic glaciers cover about 11% of the country and the focus area of this study is Vatnajökull, Iceland's largest glacier with

an area of more than 8000 km$^2$ (Figure 1) (e.g. Björnsson and Pálsson, 2008).

In this study we want to explore what impact dust events in Iceland have on the glacier surface albedo, how often they occur and what their impact on the energy balance of glaciers in Iceland is. To answer these questions, dust deposition rates calculated with a dispersion model were compared with albedo measurements on an Icelandic glacier.

## 2. Methods

### 2.1. Dust transport modelling

A recently developed scheme for dust mobilization, called FLEXDUST (Groot Zwaaftink et al., submitted) is used to estimate dust emission. The model can be applied globally, but in this study we only included dust emission from Icelandic sources. FLEXDUST produces dust emission estimates that

can be imported directly into the Lagrangian particle dispersion model FLEXPART (Stohl et al., 1998, 2005) to estimate mineral dust transport, concentrations in the atmosphere and deposition on global and regional scales. FLEXDUST is based on meteorological data from the European Centre for Medium-Range Weather Forecasts (ECMWF), land cover data by the Global Land Cover by National Mapping Organizations (GLCNMO) and additionally, for Iceland, a high-resolution (~1 arcsec) land

cover data set that identifies sandy deserts is used (Dagsson-Waldhauserova et al., 2014; Arnalds, 2015). In FLEXDUST, dust can be emitted in regions where mineral dust is available according to the land cover data set. Snow cover inhibits the dust emission. Dust emission is initiated in regions with erodible materials if a threshold friction velocity is exceeded. Contrary to the standard version of FLEXDUST, for this study dust mobilization was assumed not to be influenced by soil moisture, but it

is inhibited in case of precipitation. For the dust sources in Iceland, that are sediments rather than soils, this appears to be a better approach. We further used a combination of the erosion classes described by Arnalds et al. (2010) and the threshold values observed by Arnalds et al. (2001) to estimate threshold friction velocity. Once mobilization thresholds are exceeded, dust emission rates



are calculated following Marticonera and Bergametti (1995). It is assumed that emitted dust particles
have a size between 0.2 and 18.2 μm and follow a size distribution after Kok (2011). Dust emission
rates were calculated on a grid with 0.1˚x0.1˚ resolution for Iceland, and with a time resolution of 3
hours.

Using the dust emission rates provided by FLEXDUST, dispersion of the dust in the atmosphere was
simulated with FLEXPART version 10. Our simulations were driven with ECMWF operational analysis
data with a resolution of 1˚x1˚ globally and a nest over Iceland with 0.2˚x0.2˚ resolution. FLEXPART
simulates dispersion by transporting particles using both resolved winds and stochastic motions
representing turbulence. Dust was carried in 10 size classes and was subject to both wet and dry
removal. Further details about dust simulations with FLEXPART are provided by Groot Zwaaftink et al.
(submitted). In the current study, dust concentrations and depositions during so-called dust events,
i.e., events with strong dust deposition on Vatnajökull simulated by FLEXPART, were analysed. A
minimum modelled concentration of 6 μg m$^{-3}$ over at least two days was defined as dust event. In
particular, we studied simulated dust events at two automatic weather stations (AWS) situated on
Brúarjökull outlet (NE Vatnajökull, Figure 1), namely station B13 at ~1210 m a.s.l. and station B16 at
~1525 m a.s.l.

2.2. Measurements

For this paper, we chose the year 2012, which was characterized by warm temperatures and
exceptionally low glacier albedo on Brúarjökull. This year was not directly influenced by dust
deposition from volcanic eruptions, and albedo data from weather stations were available. We used
dust measurements in snow for the year 2013, since no measurements were available for 2012, and
compared them for the same time period (until October 2013) with the simulated spacial dust
distribution over Vatnajökull by FLEXPART.

Since 1996, AWS B13 and B16 at Brúarjökull have been used to measure the incoming ($Q_i$) and
outgoing ($Q_o$) solar radiation, incoming ($I_i$) and outgoing ($I_o$) longwave radiation, wind direction, wind
speed, air temperature and relative humidity at 2 m elevation above the surface (Guðmundsson et
al., 2006). Albedo is estimated from measured incoming and reflected short wave radiation as
$\alpha = Q_o/Q_i$. Daily albedo values were calculated as the average over 10 minute data obtained
between 13 and 14 UTC, when the solar zenith angle is smallest.

The AWS data, specifically albedo, temperature and wind, were compared with dust concentration
and deposition values from FLEXPART as well as MODIS images for the measurement period in the
year 2012 (dates in this paper are given as *days of the year* or *DOY* between DOY 130 and 283).



Surface snow samples, from the previous year's melted out firn layer, were collected in October 2013 at 16 sites on Vatnajökull (Dragosics et al. 2016). The samples contain dust deposited at these sites during the summer of 2013. The top ~8 cm of snow including impurities were brought to the laboratory, where they were melted, evaporated and the mass of the dust was weighed.

Additionally, two ~8 m long firn cores including dust layers from Brúarjökull (NE Vatnajökull), were drilled at B15 in 2015. The dust layers in the cores were dated depending on their depth and compared with mass balance measurements ($h_w \times \rho_w = h_f \times \rho_f$; where $h_w$ is mass balance given as thickness of water, $\rho_w$ is the density of water, $h_f$ is the thickness of a firn layer and $\rho_f$ is the density of firn). Dust deposition rates were estimated by measuring the mass of the dust content in the annual

layers, and compared to model results (Table 1).

### 2.3. Surface energy balance calculations

The total energy balance ($M$) for a melting glacier surface is expressed as

$$M = R + H + Hp, \qquad (1)$$

where $R = Q_i(1 - \alpha) + I_i - I_o$ is the net radiation obtained from the observed shortwave and

longwave radiation components, and $H = H_d + H_l$ is the net turbulent flux of sensible ($H_d$) and latent ($H_l$) heat calculated from the observed temperature, humidity and wind speed within the boundary layer. A one-level model with stability factor and different roughness lengths for wind-speed, temperature and humidity, described in Guðmundsson et al. (2009) was used to calculate $H_d$ and $H_l$. Heat supplied by precipitation ($H_p$) is considered negligible and the melt (ablation) $m$ is

calculated as

$$m = \begin{cases} \frac{M}{\rho_w L_f}; M \geq 0 \\ 0; \quad M < 0 \end{cases} \qquad (2)$$

where $L_f$ is the latent heat of fusion ($L_f = 3.34 \; 10^5$ J kg$^{-1}$) and $\rho_w$ the density of water (1000 kg m$^{-3}$) (e.g. Guðmundsson et al., 2006).

Albedo is a key variable in the surface energy balance and used to calculate melting. If the energy

balance is positive, this indicates an energy gain to the surface; if it is negative, it means an energy loss. The accuracy of the instruments (Kipp & Zonen CNR1) measuring longwave and shortwave radiation fluxes at AWSs was 3% (Guðmundsson et al., 2009).

To quantify the enhanced melt rates due to dust on the surface, the development of surface albedo for a dust free surface must be estimated at specific locations and meteorological conditions. This

albedo estimate and in situ AWS data may then be used to calculate the energy balance at the AWS



sites. The results can be compared to energy balance calculated from only the AWS data including the observed albedo. The development of surface albedo of snow is depending on meteorological processes in the surface boundary layer, the energy budget of the surface, snowfall events etc. A climate model that includes modelling of the boundary layer meteorology and surface albedo can

with forced data from a general circulation model be used to simulate clean surface albedo. Here we use the HIRHAM5 climate model. HIRHAM5 combines the dynamical core of the HIRLAM7 numerical forecasting model (Eerola, 2006) with the physical schemes from the ECHAM5 general circulation model (Roeckner et al., 2003). Model simulations have been validated over Greenland using AWS and ice core data (e.g. Lucas-Picher et al., 2012; Langen et al., 2015). Using the same method described in

Langen et al. (2015), we run the surface scheme in HIRHAM5 by forcing it with atmospheric parameters from a previous model run. This method allows us to implement an improved albedo scheme (Nielsen-Englyst, 2015) without running the full model. This is described in more detail in the appendix and Schmidt et al. (in preparation).

### 2.3.1. Evaluation of modelled albedo by HIRHAM5

As there was no ice at the surface at either of the two AWS's, we allowed the modelled clean surface albedo to drop to the value of clean firn, which we assumed to be 0.55. This value is based on the recommended value by Cuffey and Paterson (2010), but also represented in observed albedo in the years 2002, 2009 and 2014 (Appendix Fig. A1). For those years, measured albedo remained mostly above 0.55 for the whole measuring period. Under dry conditions the modelled albedo can only drop

to 0.77. The albedo of fresh snow was assumed to be 0.9. Based on albedo measurements this value seemed realistic after new snow events as seen in Fig. A1 in the Appendix. Sometimes measured albedo values especially in autumn can reach high values, even above 1. This can be explained due to the high solar zenith angle, multiple reflections and instrumental error (Kipp&Zonen).

The time scale $\tau_m$, which determines how fast the albedo reaches its minimum value, was chosen to

be 4 days, as it gives the best fit with the measurements without dropping below the measured values. In addition, this value gave the best fit when comparing with albedo measurements for other years with higher albedo (Appendix, Fig. A1 and Fig. A2), where the rate of the albedo decreased after a snow fall seemed realistic. Measured albedo might drop faster after a new snow event than predicted by the HIRHAM5 model, because light is penetrating through the new snow and might

reach a dust surface below.

The AWS B16 is situated in the accumulation area, but B13 is close to the equilibrium line of the glacier. This means that only in some years, as e.g. in 1997, 2004, 2005 and 2012 (Appendix Fig. A3) the mass balance was negative and the previous years' surface melted out at B13 and exposed firn




(not ice) with dust. Since 2012 was one of these extreme years, not only deposition during dust

events influenced the albedo and energy balance. At station B13 between days 206 and 225 simulation values have been manually set to the minimum value of 0.55 because HIRHAM5 simulated a snowfall event, which was not observed.

Generally, the model captures the measured albedo variability; however, the observed albedo is more variable and reaches lower values between events of snowfall. Since this is a simple model, we

are not expecting the model to capture all details. The statistical fit for HIRHAM5 compared to the AWS data showed a better fit for years with higher albedos where the previous summer surface did not melt out. The average bias, taken as the difference between HIRHAM5 and AWS data, is 0.08 for the years 1997-2014 whereas for the year 2012 it is 0.18 which means an overestimate by the model. The correlation coefficient for measured and simulated albedo data for the year 2012 is 0.77, which

is higher than the average value for other years of 0.68.

3. Results

3.1. Spatial distribution of dust deposition 2013 and total deposition rates 2012 and 2013

The annual dust deposition distribution for the surface of Vatnajökull for 2013 showed a similar pattern in the model simulation and in the observations (Figure 2). The model simulated the highest

concentrations in the south western part of Vatnajökull (Tungnaárjökull, Skaftárjökull, Síðujökull), followed by the north western and northern parts (Brúarjökull). This distribution is due to the major dust mobilization areas around Vatnajökull, such as Dynjgusandur, Tungná- and Skaftáöræfi (the area with severe erosion SW of Vatnajökull (Figure 1)), as well as the prevailing winds. The measurements of Dragosics et al. (2016) are shown as circles superimposed upon the modelled dust distribution in

Figure 2. The average dust deposition for the 16 measurement locations was 2 g m$^{-2}$. The standard deviation of the measurements, 4 g m$^{-2}$, was quite high due to one outlier with a deposition value of 16.6 g m$^{-2}$ in the SW on Tungnaárjökull. The average modelled deposition for the same locations as in the measurements is 6 g m$^{-2}$, with a standard deviation of 1 g m$^{-2}$. Thus, the model overestimated measured dust deposition by a factor of three and generated smaller dust variability. The latter was

not surprising, given the relatively coarse resolution of the model compared to the point measurements. Furthermore, variability in observed dust amounts was not only caused by the patterns of dust deposition on the glacier, but also due to windblown transport over an undulating surface, or surface melt streams washing away surface dust. Such processes were not accounted for in the modelled dust patterns. Regarding the mean concentrations, at least part of the model high

bias may, in fact, be due to a location bias in the measurements. Most of the measurement locations are in the accumulation zone of the glacier, whereas model grid cells where the measurements are



located often extend to the glacier edges where deposition amounts are higher. Regardless of whether the model bias can be explained or not, the comparison shows that the order of magnitude of dust deposition on Vatnajökull is captured by the model.

In Table 1, the measured and modelled dust deposition during the years 2012 and 2013 for stations on Brúarjökull, our main area of investigation, were reported. Again, the model tended to overestimate dust deposition; however, the order of magnitude was captured correctly.

### 3.2. Dust events on Brúarjökull 2012

FLEXPART results for both dust concentrations in the air and dust deposition on the glacier surface

were reported for the dust events for the year 2012 at station B13 (Table 2) and B16 (Table 3). Albedo, temperature and wind at 2 m elevation were measured at the AWSs, while precipitation data were taken from the ECMWF model. At station B13 there were ten modelled dust events during the measuring period (9 May to 14 October 2012), and all of them were associated with an observed albedo drop during the event at the AWS. Four events had high dust concentrations and depositions

(bold in Table 2), and six smaller events occurred as well (Figure 3). The highest deposition values were simulated during event 6 with 6.6 g m$^{-2}$ of dust deposited during a period of 14 days with an albedo drop of 0.65 from the maximum to the minimum albedo value during that period. In contrast, at station B16 (Table 3) the largest deposition (5.2 g m$^{-2}$) occurred during event 1 with an albedo drop of 0.17. Two events, one occurred during sub-freezing temperatures, and the other during

melting temperatures, were described in detail in section 3.2.1.

The albedo was almost always lower at site B13 than at site B16, due to the lower elevation and thus higher temperatures and increased melting at this site, and probably also because of its proximity to a major dust source area (Dyngjusandur). The biggest dust events happened in spring (mid-May) and autumn (end of August and October), especially at station B16. Dust event 5 coincided with warm

summer temperatures and exposure of the ablation area, where albedo at B13 reached its lowest value, 0.08, on day 223. At the lower elevation site B13 (~1210 m a.s.l.), dust deposition and concentration values during dust events were always larger than at the higher site B16 (~1525 m a.s.l.), except for event 1 (section 3.2.1). The duration of the events was also often longer at B13 than at B16. Furthermore, no dust was simulated at station B16 during event 8 (Table 3).

### 3.2.1. Case studies

Two dust events have been chosen for a detailed description. Event no. 1 (Figure 4) was by far the biggest event at B16 and temperatures were below freezing all the time, and event no. 2 (Figure 5)





happened, as was often the case, during melting temperatures. The analysis of event no. 2 was supported by the availability of a clear-sky MODIS image showing the dust cloud and deposition.

### 3.2.1.1. Dust event 1

Dust event 1 is one of four major modelled dust storms on Brúarjökull in 2012 (Figure 4) and the only event for which total simulated dust deposition was higher at station B16 (3.7 g m$^{-2}$) than at B13 (2.6 g m$^{-2}$). This explains why the albedo reached a lower value between day 134 and 139 at B16 than B13, which is very atypical. During the event, albedo dropped by 0.15 from 0.9 to 0.75 at B13 (Table 2) and by 0.17 from 0.88 to 0.72 at B16. Albedo peaked on day 133 at B16 and on day 134 at B13 because of snow fall. Simulated dust deposition started on day 134 at midday and lasted until day 136 (afternoon). This was the largest wet deposition event at both stations. At B13 (B16) there were 1.6 g m$^{-2}$ (1.3 g m$^{-2}$) dust deposited as dry deposition and 2.1 g m$^{-2}$ (3.9 g m$^{-2}$) as wet deposition, which at B16 was by far the largest deposition in a single event.

Near-surface dust concentration reached values of 193 µg m$^{-3}$ at B13 and 121 µg m$^{-3}$ at B16. Temperature decreased during the event and remained well below the freezing point, excluding the possibility that melt processes were responsible for the albedo drop. This strongly supports our hypothesis that the dust deposition caused the albedo reduction. Since dust was deposited during snowfall, the albedo drop is probably smaller than if the dust were deposited entirely by dry deposition. In fact, normally albedo increases during snowfall, so the dust deposition must have more than compensated this effect. Wind was blowing from the north during days 134-138, with high wind speeds on day 134 and 135 (B16 16 m s$^{-1}$, B13 11 m s$^{-1}$), indicating that dust was transported most likely from Dyngjusandur.

### 3.2.1.2. Dust event 2

Dust event 2 is the second largest modelled dust event in terms of dust concentration and fourth biggest in terms of total deposition at station B13 (2.5 g m$^{-2}$) but it was much smaller at B16 (0.1 g m$^{-2}$) and started later (day 146). Dust concentrations at B13 (B16) reached 225 µg m$^{-3}$ (19 µg m$^{-3}$). Dust deposition started in the afternoon on day 145 and albedo dropped on day 146 (from 0.73 to 0.60). During the whole dust event albedo dropped by 0.36 (0.86 to 0.5) at B13 and by 0.28 (0.87 to 0.59) at B16. Temperature rose above the freezing point on day 143 and this may partly explain the albedo reduction. However, the strongest albedo reduction coincided closely with the time period of the dust deposition. In particular, notice that the albedo did not decrease significantly after the end of the deposition event, even though temperatures (at least during daytime) remained above the freezing point.



290 Notice also that the albedo reduction was stronger at B13 than at B16, in agreement with the higher dust deposition at B13. Precipitation occurred until day 146, so mainly before dust deposition, suggesting that dust deposition was the main factor in this albedo drop. Wind was strongest on day 146 (13.6 m s$^{-1}$ at B13) and from SW (glacier wind), but changed to WNW until day 149.

### 3.2.2. Average dust event at B13 in 2012

295 Using the values reported in Table 2, we calculated averages to characterize an *average* dust event at the B13 site. On average a dust event at station B13 in 2012 lasted for 6 days, had a maximum dust concentration of 122 μg m$^{-3}$ and a total deposition of 2 g m$^{-2}$. Dry deposition in all cases except the first event exceeded wet deposition. This is due to the proximity of the measurement site to the source area and gravitational settling of larger particles, which dominated the removal near the 300 source. The albedo is on average lowered by 0.18 in a dust event. This large reduction had a strong impact on the radiation and energy balance of the glacier. The average temperature during dust events was -2°C (at ~1210 m elevation) and the prevailing wind direction in 80% of the events was northerly, in 20% it is SW (the direction of the glacier wind on Brúarjökull). Average ECMWF precipitation during events was ~23 mm.

### 3.3. Surface energy balance impact of dust deposition

Precipitation of dark dust particles on a glacier surface lowers the surface albedo, thus also the surface energy balance and in general increases the energy available for melt. In order to estimate the contribution of this effect, the surface energy balance (and the surface melt from energy balance) at the two AWS sites B13 and B16 was estimated from the AWS data in 2012. To estimate 310 the effect, the regional climate model HIRHAM5 was used to simulate a clean glacier surface for the weather conditions occurring at the AWS B13 and B16 in 2012. The simulated clean surface albedo (black line in Figure 7) is compared to the observed albedo including impurities (red line in Figure 7).

The difference between the modelled clean surface and the real surface is greater at B13 than B16. This was expected since dust concentration is much higher at the lower site B13 and snowfall more 315 common at the upper site B16. We also know from mass balance measurements, that at B13 all the winter snow melted, exposing firn and surface dust from previous years (this happened at day ~205). With addition of dust from dust events starting on days 202 and 220 (Figure 8) the albedo values dropped very low at B13 between days 220 and 236. The simulated energy balance did not predict the snow from the previous winter to have been melted away completely, exposing the firn layer.

High temperatures at B13 up to ~5 °C coincide with dust event 5, which caused peaks in snow melt of 8.13 cm w.e. d$^{-1}$ on day 222. In the autumn, after day ~240, the energy balance was mostly negative.



Low net radiation (due to low solar radiation and high albedo caused by snow fall) accompanied with negative turbulent heat fluxes (due to air temperatures below zero and strong winds) resulted in negative total energy, i.e. no energy available for melting in 2012.

The total summer melt at B13 in 2012 estimated from the energy balance calculated for a dust free surface was 1.7 m w.e., whereas for the measured albedo the melt was estimated at 2.8 m w.e. From this we conclude that the melt increased by 1.1 m w.e., or by ~60%, due to dust deposition, and melting out of the dusty firn surface below. Other impurities such as black carbon or organic material cannot be excluded to affect the albedo, but their contribution was expected to be very

small/negligible (from in-situ investigation during our decade long early-, mid- and late-summer visits to the ice cap). At the higher site, B16, 1.0 m w.e. of snow melt was calculated for the modelled dust free surface and 1.6 m w.e. when using the measured albedo, which results in 0.6 m more snow melt caused by dust on the surface. The increase in melt is similar to that in B13, i.e. additional 60%.

4. Discussion and conclusion

In this paper, we have shown that dust events modelled by FLEXDUST correspond to reductions in the observed albedo at two AWS sites on Vatnajökull. This indicates that the model is able to capture the occurrence of individual dust events. Furthermore, we showed that the model captures both the observed spatial distribution of dust on the glacier as well as the magnitude of the total annual deposition amounts. This suggests that the model can be used for longer-term studies, to quantify

the dust deposition on Vatnajökull, including its interannual variability. Table 2 shows the dust events of the year 2012 at station B13 on Brúarjökull, where in total 10 dust events occurred; four main events and six smaller events. The AWS measurements show a drop in albedo in connection to all dust events predicted by FLEXPART within the AWS's survey period. The prevailing wind direction during dust events at site B13 is from a northerly direction, while for the whole period downslope

(SW) winds dominate. The wind direction during dust events corresponds to the main dust source Dyngjusandur, north of Vatnajökull. At site B16, situated further upglacier, 9 dust events occurred (Table 3) where the first dust event with ~5 g m$^{-2}$ of dust deposited within 3 days was by far the largest.

In Arnalds et al. (2014) average deposition of dust on Icelandic glaciers is estimated as ~400 g m$^{-2}$ yr$^{-1}$

which seems to be overestimated. Their estimate includes periodic tephra deposition and large dust events based on a country average and it does not adequately account for topographic differences and that much of the glacial areas are upwind for dry winds from the main dust sources at the glacial margins. With FLEXPART, we calculated much lower annual deposition rates for Vatnajökull and its surroundings in 2013 (Figure 2), up to 34 g m$^{-2}$ in the SW of the glacier. Moreover, modelled values



for dust deposition rates on Brúarjökull of 20 g m$^{-2}$ (B13) and 10 g m$^{-2}$ (B16) for 2012 were much
      lower.

      Firn core B drilled on Brúarjökull showed a dust layer of ~8 g m$^{-2}$ for 2012 (Table 1), in very good
      agreement with the simulated dust of 8.5 g m$^{-2}$. At firn core (A), drilled in the immediate vicinity of
      core B, observed deposition rate was much smaller (1.7 g m$^{-2}$), showing the large spatial variability

and consequent uncertainty in comparing point measurements to model simulations. We thus
      consider the model results satisfactory if they are in the same order of magnitude as observed dust
      amounts in ice cores or snow samples.

      To estimate the impact of dust on the surface energy balance and melt rates, the regional climate
      model HIRHAM5 was used to simulate the surface albedo for a dust free, i.e. clean snow surface

during the summer 2012. The surface energy balance (and melt rate) was calculated using the
      simulated albedo and the albedo observed from the AWS data. At the lower site, B13, the difference
      between dust free and real surface is 1.1 m w.e. of more snow melt (1.7 m w.e. snow melt for the
      clean surface and 2.8 m w.e. for the real surface). This does not only include dust events lowering
      surface albedo, but also dust and tephra that was deposited during previous years melting out from

below. At the supper site B16 the difference results in 0.6 m of more snow melt (1.0 m w.e. for the
      clean surface and 1.6 m w.e. for the AWS). Since B16 is situated in the accumulation area, no dust
      expected to melt out from below. It cannot be excluded that small amounts of organic material or
      black carbon are deposited on the snow surface and influence albedo, but from in situ investigations
      this has not been observed in this area.

The year 2012 was a year of intensive summer melt. At site B13 on Vatnajökull the measured
      summer mass balance was 2.3 m w.e. mass loss, which means 0.5 m more mass loss than the
      average since 1993 (1.7 m w.e.). Summer mass balance measurements on Vatnajökull show 2.3 m
      w.e. of total mass loss at B13 which is 0.5 m less melt compared to calculated energy balance
      converted into snow melt (2.8 m w.e.). Most of these differences is assigned to summer snow fall

that melts, and was not captured with the mass balance measurements.

      Oerlemans et al. (2009) reported that decreased albedo at Vadret da Morteratsch glacier caused an
      additional removal of about 3.5 m of ice for the 4 year period 2003–06. This means 0.9 m more melt
      on average per year. Gabbi et al. (2015) compared a glacier surface with deposits of black carbon and
      Saharan dust to pure snow conditions for a 100 year period (1914-2014). They found that the mean

annual albedo decreased by 0.04–0.06, therefore the mean annual mass balance was reduced by
      about 28–49 cm. These alpine melt rates due to impurities are in the same order of magnitude as our
      results.


Albedo comparisons for other years (Appendix, Fig. A3) have shown very low albedo values for the
years 1997, 2004, 2005 and 2012. The surface dirt causing the low albedo in 1997 is related to the

Grímsvötn eruption in 1996, and the following huge jökulhlaup with deposition of fine grained
particles on Skeiðarársandur sandur plain.  This was a vast source of dust in the dry and warm 1997
summer. The low albedo in 2005 and 2012 most likely also related to the 2004 and 2011 Grímsvötn
eruptions (e.g. Guðmundsson et al. 2004, Möller et al. 2013.) In 2004 increased melt rates due to
high wind-driven turbulent heat fluxes in the end of July followed by exceptionally warm and sunny

weather in August sped up melting into old firn (Guðmundsson et al. 2006).

The results in this paper shows positive radiative forcing impact on snow melt of Icelandic glaciers
caused by deposition of dust that strongly enhances absorption of light. The duration of dust
radiative effects on glacier surfaces is extended compared to purely atmospheric effects because of
the short lifetime of dust in the atmosphere.


Acknowledgements

The study described in this manuscript was supported by NordForsk as part of the two Nordic
Centres' of Excellence Cryosphere-atmosphere interactions in a changing Arctic climate (CRAICC),
and eScience Tools for Investigating Climate Change (eSTICC). Part of this work was supported by the

Centre of Excellence in Atmospheric Science funded by the Finnish Academy of Sciences Excellence
(project no. 272041), by the Finnish Academy of Sciences project A4 (contract 254195). Data from in
situ mass balance surveys and on glacier automatic weather stations is from joint projects of the
National Power Company and the Glaciology group of the Institute of Earth Science, University of
Iceland. C. Groot Zwaaftink was also funded by the Swiss National Science Foundation SNF (155294),

and Louise Steffensen-Schmidt, Finnur Pálsson and Sverrir Guðmundsson by the Icelandic Research
Fund (project SAMAR) and the National Power Company of Iceland. Ólafur Arnalds was in part
funded by Icelandic Research Fund (grant no. 152248-051)

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





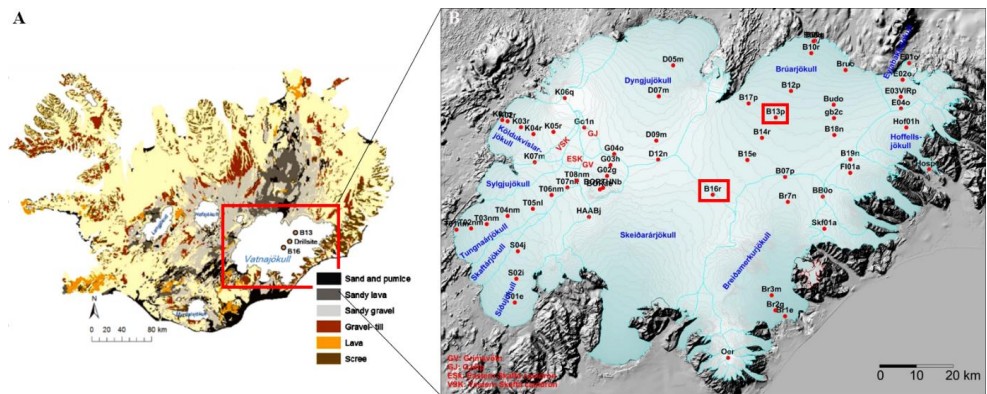

Figure 1. A) Iceland with glacier outline and soil map adapted from Arnalds (2015). B) Vatnajökull ice cap and mass balance survey sites (e.g. Björnsson et al. 2013). Ice divides are shown with light blue lines. The two AWSs at B13 and B16 on Brúarjökull are highlighted with red frames. Vatnajökull map by Glaciology group, Institute of Earth Sciences, University of Iceland.


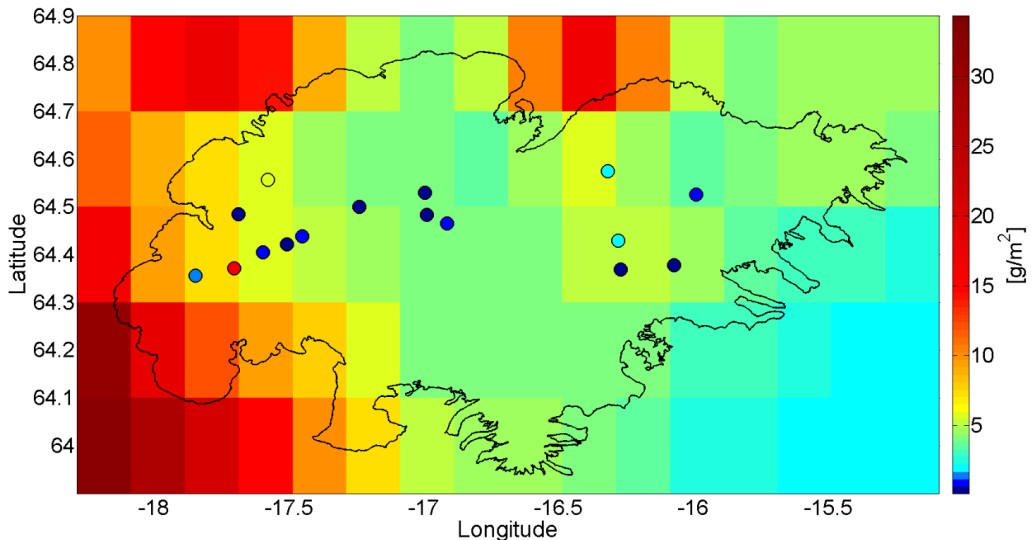

Figure 2: FLEXPART model simulation of the spatial dust distribution on Vatnajökull during 2013. The circles show the location of snow sample sites with dust deposition for the same year.



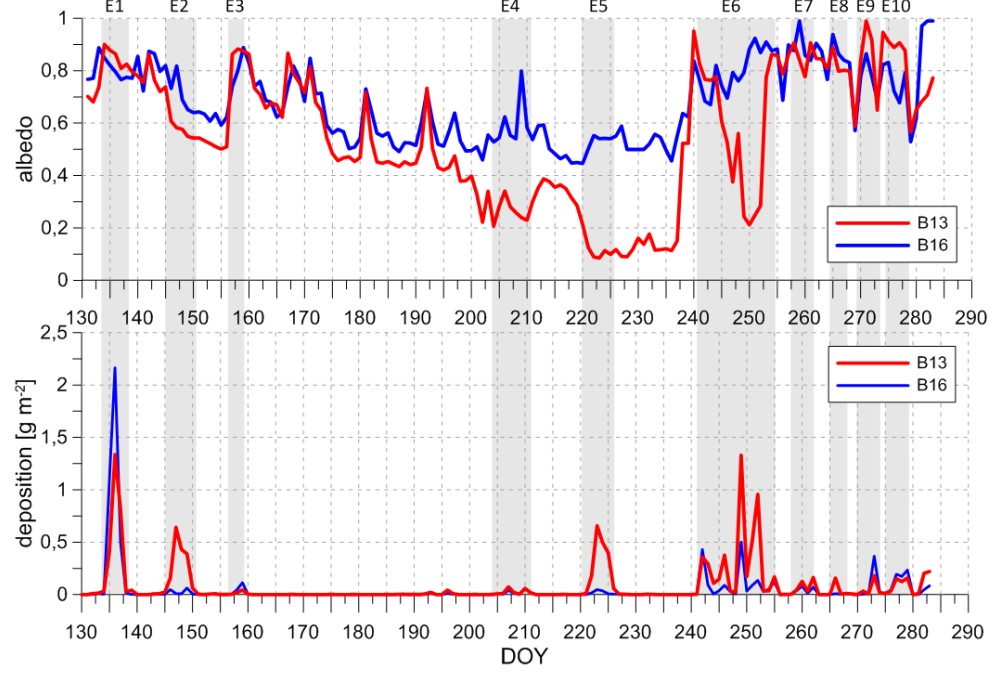


*Figure 3: Upper graph: Albedo measurement from the AWS at B13 in red and B16 in blue for the measurement period in 2012. Lower graph: Daily dust deposition showing dust events modelled by FLEXPART. Dust events are highlighted in grey and named E1-E10.*

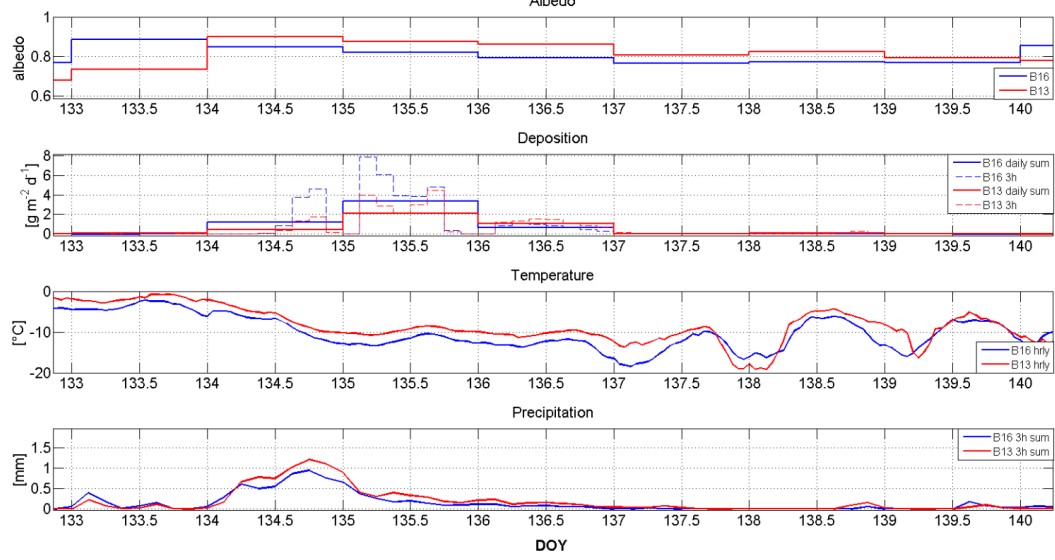

550       *Figure 4: Observed albedo, simulated dust deposition, observed temperature and simulated precipitation dust event no. 1 at stations B16 (blue) and B13 (red). Modelled deposition is shown for 3-hourly and daily averages.*





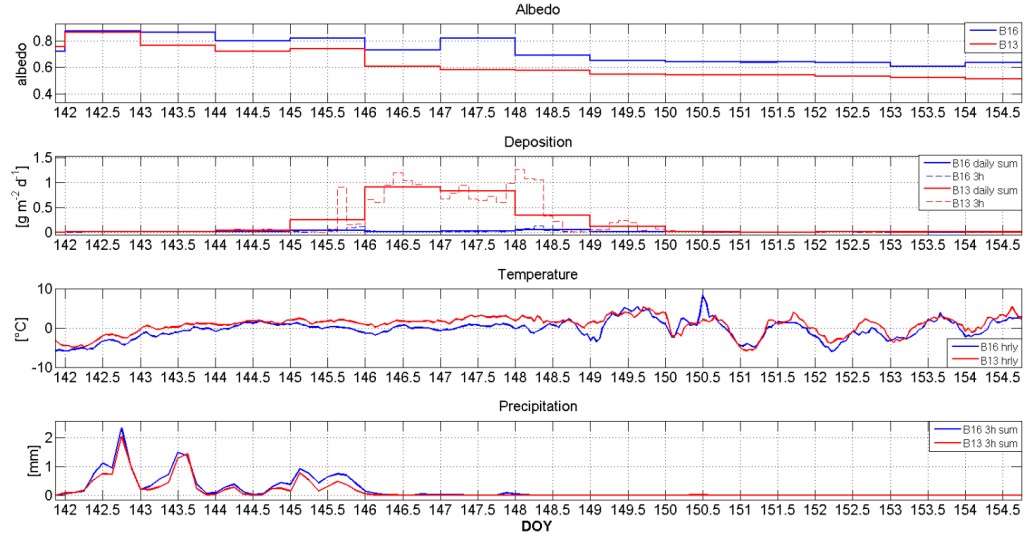

*Figure 5: Observed albedo, simulated dust deposition, observed temperature and simulated precipitation dust event no. 2 at stations B16 (blue) and B13 (red). Modelled deposition is shown for 3-hourly and daily averages.*


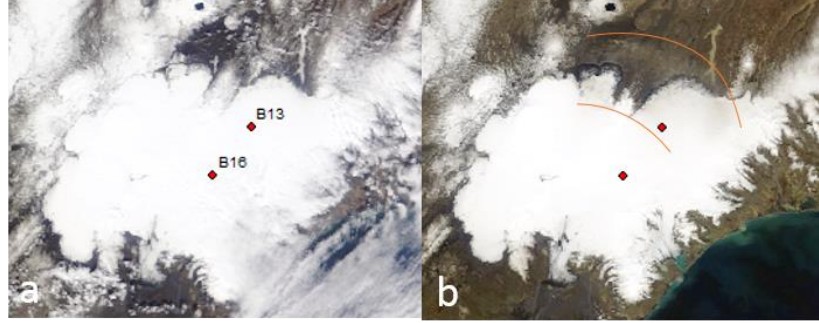

*Figure 6: MODIS images of Iceland on a) 20 May 2012 (day 141) and b) 28 May 2012 (day 149). Notice the brownish hues (between orange lines) on Brúarjökull outlet (north-Vatnajökull) after the dust event, which indicate that dust was deposited on the glacier. Image courtesy of MODIS Rapid Response System at NASA/GSFC. http://rapidfire.sci.gsfc.nasa.gov/*





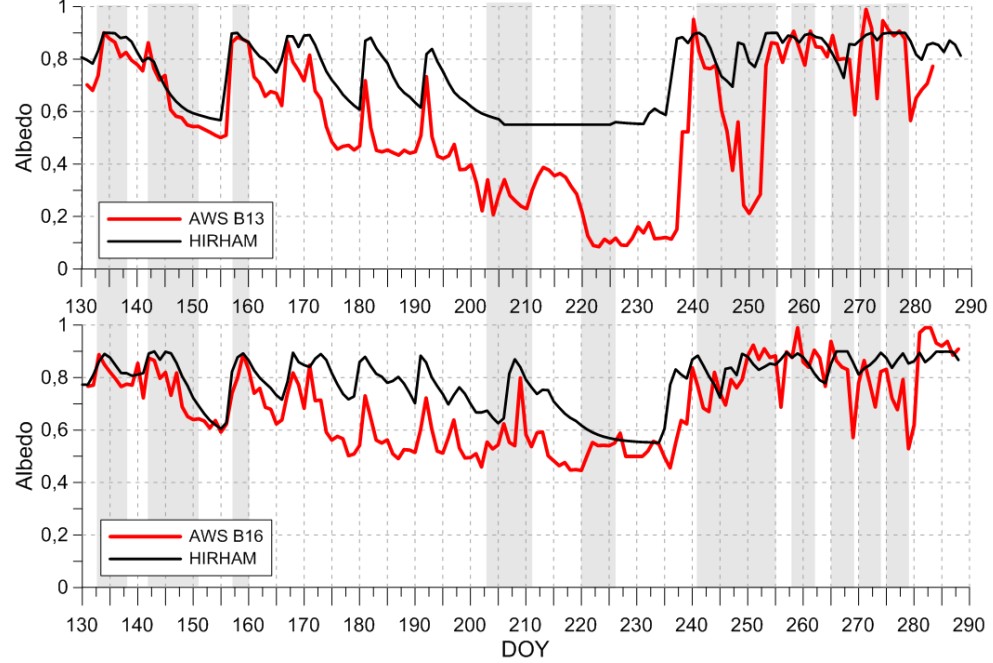


*Figure 7: Measured albedo (red line) and albedo simulated with HIRHAM5 (black line) for a clean glacier surface without dust at the stations B13 (upper graph) and B16 (lower graph). Highlighted in grey are modelled dust event periods by FLEXPART.*




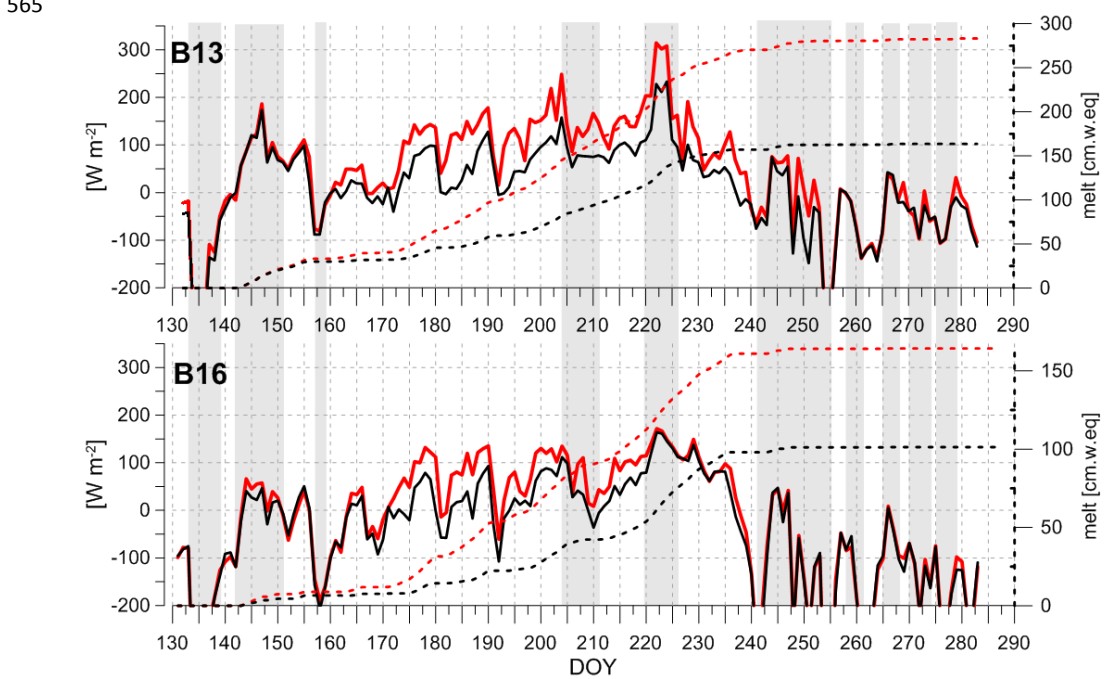

*Figure 8: Measured energy balance (red line) and energy balance with simulated albedo with HIRHAM5 (black line) for a clean glacier surface without dust at the stations B13 (upper graph) and B16 (lower graph). Cumulative snow melt is shown in dotted lines for AWS in red and HIRHAM5 in black. Highlighted in grey are modelled dust event periods by FLEXPART.*






*Table 1: Total dust deposition [g m$^{-2}$] at stations on Brúarjökull in 2012 and 2013. Drill site A (Figure 1) is situated at station B15, drill site B 600 m below B15 at 1400 m elevation.*

| 2012 | Measurements | Model |
|---|---|---|
| B16 | | 10.4 |
| B13 | | 20.5 |
| firn core 2015 A | 1.7 | 9.1 |
| firn core 2015 B | 7.9 | 8.5 |
| **2013** | | |
| B13 | 2.0 | 9.4 |



*Table 2: Dust events at station B13. Reported are the modelled maximum and minimum dust concentration, the maximum simulated daily deposition as well as the total deposition during the event, the measured albedo change, maximum and minimum temperature and wind direction from the AWS, and the precipitation sum from the ECMWF model.*

| | | | Model | | | AWS | | | | | Precipitation ECMWF [mm] |
| Event Nr. | DOY | Duration [days] | Concentration [µgm⁻³] | Deposition [gm⁻²] | | Albedo change | | Temperature [°C] | | Wind | |
| | | | max | max | sum | max-min | start-end | min | max | main direction | sum |
| 1 | **133-138** | 6 | 192,84 | 2,09 | **3,70** | 0,15 | 0,15 | -12,7 | -4,8 | N | 31 |
| 2 | **142-150** | 9 | 225,12 | 0,90 | **2,48** | **0,36** | 0,36 | -2,9 | 3,4 | E,S to NW | 24 |
| 3 | 157-158 | 2 | 13,75 | 0,06 | 0,09 | 0,26 | 0,26 | -3,8 | -0,3 | NNE | 11 |
| 4 | 204-210 | 7 | 49,38 | 0,10 | 0,23 | 0,13 | 0,11 | -0,1 | 2,4 | S to N | 19 |
| 5 | **220-225** | 6 | 212,07 | 1,02 | **2,68** | 0,04 | 0,04 | 2,0 | 4,8 | SW | 0 |
| 6 | **241-254** | 14 | 298,29 | 1,77 | **6,60** | **0,65** | 0,09 | -6,2 | 2,4 | SW to N,SE | 114 |
| 7 | 258-261 | 4 | 44,89 | 0,19 | 0,37 | 0,13 | 0,13 | -4,9 | -2,4 | NW to N | 14 |
| 8 | 265-267 | 3 | 49,87 | 0,21 | 0,23 | 0,30 | 0,30 | -5,1 | 1,4 | SW,SE | 48 |
| 9 | 270-272 | 3 | 67,86 | 0,29 | 0,33 | 0,34 | 0,34 | -4,0 | -1,8 | W,NE,N | 32 |
| 10 | 275-278 | 4 | 67,34 | 0,22 | 0,64 | 0,35 | **0,35** | -10,3 | -2,5 | NNE | 29 |





*Table 3: Same as Table 2 but for station B16.*

| Event Nr. | DOY | Duration [days] | Model | | | AWS | | | | | Precipitation ECMWF [mm] |
|---|---|---|---|---|---|---|---|---|---|---|---|
| | | | Concentration[µgm⁻³] | Deposition [gm⁻²] | | Albedo change | | Temperature [°C] | | Wind | |
| | | | max | max | sum | max-min | start-end | min | max | main direction | sum |
| 1 | **134-136** | 3 | 120,96 | 3,36 | **5,22** | 0,17 | 0,17 | -14,3 | -6,0 | N | 25 |
| 2 | **145-149** | 6 | 19,39 | 0,05 | **0,12** | **0,28** | 0,28 | -4,1 | 2,5 | once around clockwise | 2 |
| 3 | 157-158 | 2 | 15,33 | 0,12 | **0,17** | **0,27** | 0,27 | -5,1 | -0,3 | NNE | 16 |
| 4 | 206-210 | 5 | 21,14 | 0,06 | 0,15 | 0,26 | 0,04 | 2,5 | 7,6 | N,SW,N | 4 |
| 5 | 221-223 | 3 | 15,01 | 0,07 | 0,15 | 0,01 | 0,01 | 2,2 | 2,9 | SW | 2 |
| 6 | **241-254** | 14 | 71,44 | 0,66 | **2,34** | **0,25** | -0,04 | -8,4 | 0,7 | N,SW | 110 |
| 7 | 258-259 | 2 | 27,92 | 0,07 | **0,10** | 0,13 | 0,13 | -6,8 | -3,7 | W | 4 |
| 8 | | | | | | no event | | | | | |
| 9 | 270-273 | 3 | 51,31 | 0,53 | 0,55 | 0,18 | 0,18 | -5,4 | -4,7 | SW,E,N | 22 |
| 10 | **275-278** | 4 | 45,27 | 0,22 | **0,64** | **0,30** | 0,30 | -8,7 | -4,2 | N | 14 |



# Appendix

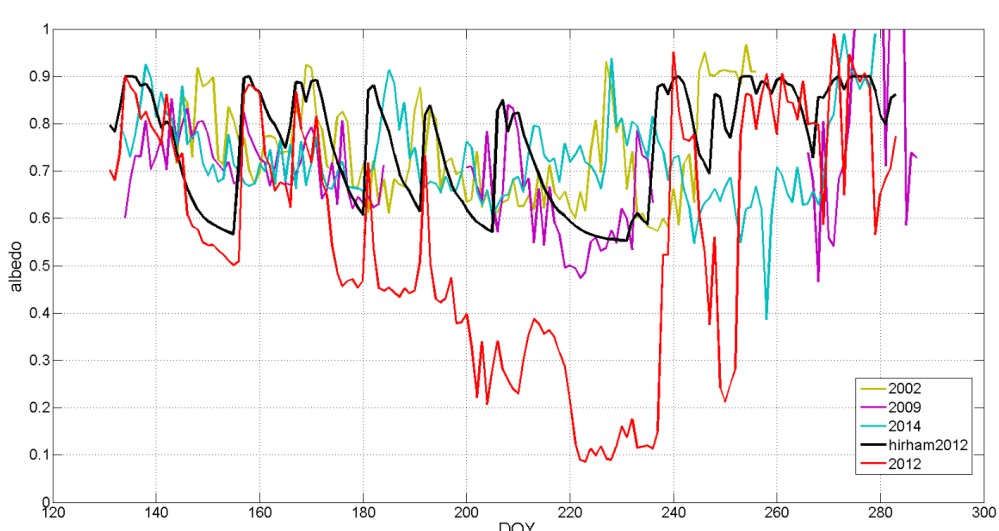

*Figure A1: Albedo measurements for the AWS station B13 for the years 2002, 2009, 2012, 2014 (2002, 2009, 2014 are years of little or none surface dust) compared to the modelled surface albedo by Hirham5 (no surface dust assumed in the model) for 2012.*

Albedo at station B13, which is close to the ELA is shown in Fig. A1. The year 2012 was modelled by HIRHAM5 (in black) and compared to the AWS (in red) and to 3 other years of albedo measurements

showing high albedo, but no melt out of the previous summer surface and a similar speed of albedo drop after a snow event. Also interesting is that after snowfall events the albedo usually peaks up to 0.9 even in the summer. If there is dust on the surface, prior to snowfall, light penetrates through the fresh thin snow cover, thus some light is absorbed by the dust and albedo will be lower than 0.9. This may help explain the low (~0.7) value of albedo in 2012 after snowfall events between days 180 and

595   200.




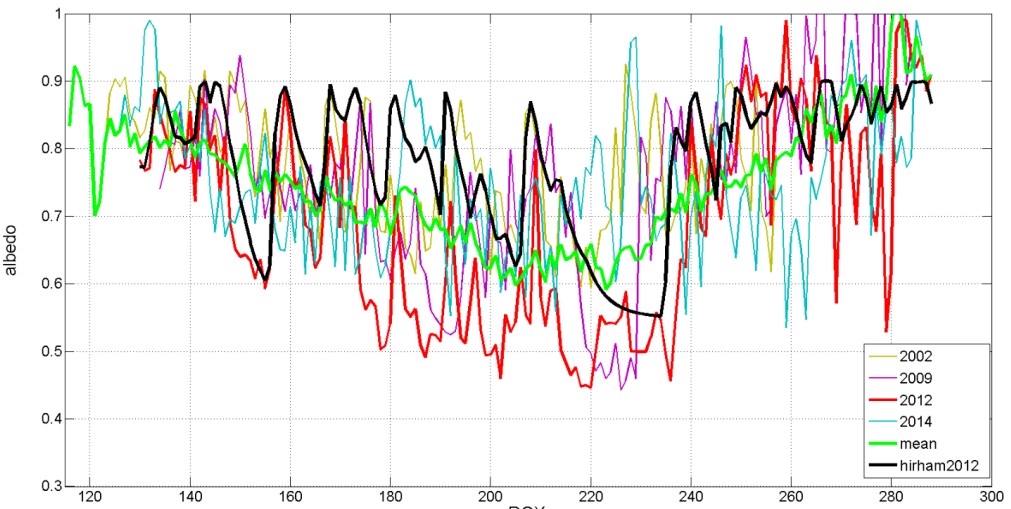

*Figure A2: Albedo measurements at the AWS station B16 in the years 2002, 2009, 2012, 2014 (years of clean surface) compared to the modelled surface albedo by Hirham5 for 2012 (black curve) and the measured albedo mean for all years since 1997 (green curve).*

The average measured albedo at B16 (Fig. A2 in green) since 1997. In black the albedo estimate from HIRHAM5 model run for 2012 is shown (no dust on surface assumed), showing similar character as the measured albedo for the other years.

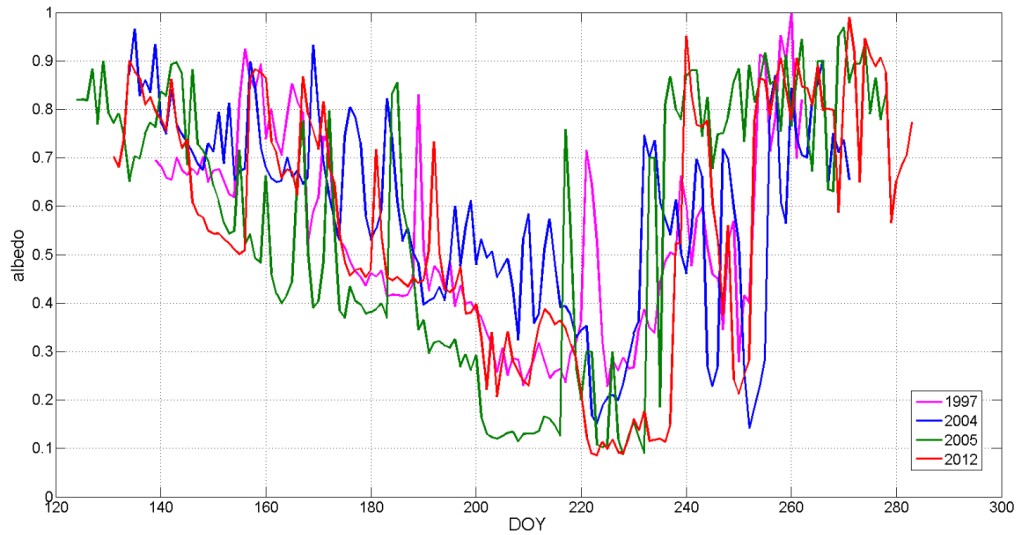

*Figure A3: Albedo measurements at the AWS station B13 for selected years of very low albedo, 1997, 2004, 2005 and 2012.*



Fig. A3 shows years with very low albedos like in 2012 (red). The surface dirt causing the low albedo in 1997 is related to the Grímsvötn eruption in 1996, and the following huge jökulhlaup with deposition of fine grained particles on Skeiðarársandur sandur plain. This was a vast source of dust in the dry and warm 1997 summer. The low albedo in 2005 and 2012 is related to the 2004 and 2011 Grímsvötn eruptions (e.g. Guðmundsson et al. 2004, Möller et al. 2013.) In 2004 increased melting

rates due high wind-driven turbulent heat fluxes in the end of July followed by exceptionally warm and sunny weather in August sped up melting into old firn (Guðmundsson et al. 2006).

- Modelling of albedo evolution in dust free conditions with HIRHAM5

HIRHAM5 combines the dynamical core of the HIRLAM7 numerical forecasting model (Eerola, 2006) with the physical schemes from the ECHAM5 general circulation model (Roeckner et al., 2003).

Model simulations have been validated over Greenland using AWS and ice core data (e.g. Lucas-Picher et al., 2012; Langen et al., 2015). Using the same method described in Langen et al. (2015), we run the surface scheme in HIRHAM5 by forcing it with atmospheric parameters from a previous model run. This method allows us to implement an improved albedo scheme (Nielsen-Englyst, 2015) without running the full model. One drawback of this method is that it neglects feedbacks between

the surface and the atmosphere. However, since we are only interested in the albedo, and the temperature of the glacier surface of Vatnajökull in the summer is typically around the melting point, the error due to neglected feedbacks is likely small. The simulated albedo was interpolated to the AWS positions using bilinear interpolation of the four nearest grid points.

The albedo parameterization is similar to that described in Oerlemans and Knap (1998), with the

albedo decaying exponentially after a fresh snow fall depending on the age of the snow at the surface. However, unlike in Oerlemans and Knap (1998), the decay rate and the minimum albedo of the model depend on the surface temperature. If the surface temperature is -2°C or lower, no melt occurs and we characterise the snow as dry.

For each model time step, the albedo is updated using

$$\alpha_{snow}^{n+1} = \left( \alpha_{snow}^n - \alpha_{\{d,m\}} \right) \exp\left( \frac{-\delta t}{\tau_{\{d,m\}}} \right) + a_{\{d,m\}} \tag{3}$$

where $\alpha^n$ is the albedo from the previous day, $\alpha_{\{d,m\}}$ is the minimum albedo, which depends on whether the snow is under dry or melt conditions; $\delta t$ is the model time step, and $\tau_{\{d,m\}}$ is a timescale which determines how fast the albedo reaches its minimum values under dry or melting conditions.

Under both melting and dry conditions, the albedo can only be refreshed to a higher value due to

snowfall. In order to take the effect of rain into account, the albedo is refreshed if a certain amount





of the total precipitation is snow. The model has a partial refreshment scheme, where the value of the albedo is refreshed and depends on the amount of snow. The refreshment rate is

$$rate = \min\left[1, \frac{S_f}{\delta t \cdot S_0}\right] \quad (4)$$

where $S_f$ is the snowfall in metres and $S_0$ is the critical snowfall in metres per time step, equal to 0.3 m, which is needed to refresh the albedo to its maximum value. The albedo will be updated at each time step using

$$\alpha_{snow}^{n+1} = \alpha_{snow}^{n} + rate \cdot (\alpha_{max} - \alpha_{snow}^{n}). \quad (5)$$

In the case of small snow depths, the surface is affected by the underlying surface and an Oerlemans-Knapp transition (Oerlemans and Knap, 1998) is used to ensure a smooth transition between snow and ice.