# Peer review of "Impact of dust deposition on the albedo of Vatnajökull ice cap, Iceland."

_The Cryosphere, 2016_

## Referee Comment (RC1) · Anonymous Referee #1 · 17 Oct 2016

Summary:

The paper by Dragosics et al. applies modeling and field observations to determine the impact of mineral dust deposition on the radiative properties of Vatnajökull's ice cap in Iceland. Field observations, gathered during 2013, and AWS data are compared with model simulations of FLEXPART and HIRHAM5 parameterized on Iceland's land cover and meteorology. The authors state that dust depositions have a strong impact on the mass balance of the ice cap, which is a novel and interesting conclusion.

General comments:

The paper is rather well structured and its hypotheses are relevant with respect to the impact of light-absorbing impurities (LAI) on the cryosphere. The referencing is quite complete and state of art knowledge is developed in the introduction.

[Figure]

The comparison between observed and simulated dust depositions showed quite big differences. For example, in Figure 2 I cannot see the "similar pattern" (line 208-209) the authors refer to. Furthermore, any direct correlation between observed and simulated data is not provided. I suggest to expand the discussion about this comparison, highlighting possible causes of the differences (e.g. the timing of the field campaigns).

The choice of the year 2012 should be strenghtened. I don't understand why the authors did not use 2013 as a test year, since they have field data for that year. If I understand well, they are comparing field observation from 2013 with simulations of 2012. In my opinion, this needs a stronger justification. Here they are assuming that the spatial patterns of dust deposition on the ice cap are equivalent from year to year.

I suggest to delete the part on MODIS data, since they are not used quantitatively in the study. In Figure 6, I cannot recognize any dust plume or deposited dust on the ice cap. If anything, other satellite (e.g. Landsat) and model products can be used to represent dust plumes and/or depositions. If you want to keep MODIS data, I suggest to compare data from the AWS with MODIS snow albedo time series (MOD10A1, MYD10A1), which could be very interesting from a remote sensing perspective.

I suggest to add in the introduction a description of the "state" of Vatnajökull ice cap (e.g. mass balance data), in order to ensure a broader impact of the paper. Why is it important to study the impact of LAI on Vatnajokull? Is there any missing link between the temperature increase and ice melting? In the introduction, reference to the impact of LAI on Greenland (e.g. Tedesco et al. 2016 TC, Dumont et al. 2014 Nat. Geo.) could also be helpful to describe the process on ice sheets, which is more interesting for climate analysis.

The Appendix is more suitable as Supplementary Information.

The dust mobilization scheme (FLEXDUST) is based on a paper that was submitted to JGR (Groot Zwaaftink et al.). I suggest to expand the description of this scheme, sinceat the moment the reader cannot have details on it.

[Figure]

I'm not an expert in mass balance modeling, so I don't have specific comments on it. In any case, the impact on mass balance is evaluated using only two AWS dataset. Is it possible to extrapolate this information to the whole Vatnajokull ice cap? or the impact of mineral dust could be limited to some areas of the glacier?

I think that the paper can be published in TC once these comments are addressed and discussed. Specific comments are reported below.

Specific comments:

Figure 1: It is very small and labels are difficult to read. I suggest to enlarge Fig.1A and to remove Fig.1B since the details are not relevant for this study. As far as I understand, only data from B13 and B16 (+ the firn core) are used in this work.

Figure 2: I suggest to use a linear scale bar palette for dust deposition.

line 56 Here briefly describe the impact of LAI on Greenland Ice sheets.

line 76: delete "want".

line 77: please rephrase this sentence.

line 92: "Snow cover inhibits the dust emission": explain better or remove this sentence

line 111: Why did you choose this treshold? Please explain.

line 116-117: here some references are needed. 2012 was an extreme year in the northern hemisphere, with strong melting in Greenland (e.g. Nghiem et al. 2012 GRL).

line 118-121: here I don't understand why you chose 2012 as a test year.

line 120: replace "spacial" with "spatial".

line 128-130: remove this part on MODIS data.

line 154: "Albedo is a key variable in the surface energy balance and used to calculate melting", change with : "Albedo is a key variable in the surface energy balance and it

is used to calculate ice melting"

line 160: delete "may then be" and replace with "it is"

line 163-165: please rephrase this sentence

line 173: I personally don't like reference to paper in preparation.

line 181: delete "seemed" and replace with "is assumed to be"

line 181- 183: did you remove these data? please explain

line 188: delete "seemed" and replace with "was"

line 188-190 Unsupported statements. Add some reference or lose the phrase

line 194: delete "(not ice)"

line 194: add more details on this point, what do you mean with "extreme year"?

line 194-195: this is a fundamental aspect. Decoupling the effect of dust and other meteoclimatic forcing is a big issue. Explain better possible influence on your estimations

line 198-205: these are results, integrate in Section 3.

line 208-209: I can't see this "similar pattern". It looks like observational points don't show a marked spatial correlation, which is present in model simulations. This is supported by the IDW map showed in Figure 6 of Dragosics et al. 2016 (Arab. J. Geosci.), which shows "bulls eye" that are typical errors of spatial interpolation of uncorrelated data. I would expect that on a large ice cap such as the Vatnajökull, dust concentration on surface snow should in theory feature some spatial structure. Probably the impact of snow falls, melting and run off can redistribute the dust concentration. Only 16 samples on a large area (8000 km2) are too few to capture these complex processes of deposition and redistribution. Please discuss these aspects in the paper.

line 226: Confused sentence. Please rephrase

line 232: delete "however, the order of magnitude was captured correctly", you already said that in line 229

line 259: I don't see the dust cloud, nor the dust deposition from Figure 6.

line 284: use "from .. to .." in both parethesis

line 306: replace "precipitation" with "deposition"

line 324: confused sentence, please explain.

line 328-331: This should be better explained. BC cannot be excluded since its impact on snow is hardly visible with naked eyes (Warren 2013 JGR).

line 370: delete the "s" from "supper"

---

## Short Comment (SC1) · 31 Oct 2016

The authors conducted both field observations and model simulations to investigate the impact of dust deposition on the albedo of Vatnajökull's ice cap. The results are very interesting and advance our understanding of dust-snow albedo effect. I have a short comment on the discussion part.

The authors did not provide enough discussions on some important factors that could influence the dust impact on reducing snow albedo. For example, previous studies found that the impact of BC/dust deposition on snow albedo reduction is significantly affected by how particles mix with snow grains and different snow grain shapes (Liou et al., 2014; He et al., 2014) as well as snow aging processes (Flanner et al., 2007). I suggest adding some discussions on this aspect.

[Figure]

References:

Liou, K. N., Takano, Y., He, C., Yang, P., Leung, L. R., Gu, Y., and Lee, W. L.: Stochastic parameterization for light absorption by internally mixed BC/dust in snow grains for application to climate models, J. Geophys. Res.-Atmos., 119, 7616-7632, doi:10.1002/2014jd021665, 2014.

He, C., Li, Q. B., Liou, K. N., Takano, Y., Gu, Y., Qi, L., Mao, Y. H., and Leung, L. R.: Black carbon radiative forcing over the Tibetan Plateau, Geophys. Res. Lett., 41, 7806-7813, doi:10.1002/2014gl062191, 2014.

Flanner, M. G., Zender, C. S., Randerson, J. T., & Rasch, P. J.: Present day climate forcing and response from black carbon in snow. J. Geophys. Res.-Atmos., 112(D11), 2007.

---

## Short Comment (SC2) · 4 Nov 2016

Dear C. He, Thank you for your comment, I will consider your suggestions.

---

## Referee Comment (RC2) · Anonymous Referee #2 · 27 Feb 2017

The main focus of this paper is on modelling dust deposition on the Vatnajökull ice cap and studying the effects of dust on the surface albedo and melting rates. Data from automatic weather stations (including albedo data) on Brúarjökull, a northern outlet from Vatnajökull, are used for control, as well as measurements of dust concentration in snow samples from 16 ice-cap sites and on two 8 m firn cores. The authors demonstrate that significant changes in albedo occur on Vatnajökull in connection with events of dust deposition. They calculate the effect of the albedo change on the energy balance on the glacier surface and conclude that the dust deposition during the summer 2012 enhanced melting by 60% as compared with (modelled) melting that would have occurred in the absence of dust. This paper is an important contribution to the study of factors affecting mass-balance of glaciers in Iceland. Publication can be recommended following amendments taking the following comments into account.

Specific comments and corrections:

Figure 2 and associated discussion on page 52: Neither the figure caption nor the text is clear on exactly which period is being displayed here for the modelling results: The glaciological year 2012-2013, the entire calendar year 2013 or the part of 2013 leading up to the sampling expedition in October 2013 (data from samples collected during that expedition are displayed along with the model results in the figure). It is probably the period JD 130-283, but this should be explicitly mentioned.

The two case studies on Dust events 1 and 2 (Figures 3-5) are well described and the authors present good reasons for focusing on those events, comparing measured albedo drops with modelled dust deposition. Since both are, however, spring events, it is a bit surprising that other events do not receive comparable scrutiny, like for example the summer event E5 during JD 220-227 (Fig. 3), or the September events after JD240.

The results presented in Fig. 8 are compelling and of great interest, indicating that the dust deposition at the two Brúarjökull sites enhances total summer melting by 60% during 2012.

L42-43 "The snow-albedo feedback, where radiation absorption is enhanced due to impurity content in snow and ice is indicated by complex processes..." Further clarification needed here, what is meant by "complex processes"?

L64-66 "Iceland is one of the most active aeolian places on Earth, even though it is not situated in an arid climate (Arnalds et al., 2016). Due to the large area of sandur plains and strong winds resulting in numerous dust events."

"aeolian place" is not well put, and second sentence is subordinate, meaning that it shouldn't stand on its own.

L212 Dynjgusandur → Dyngjusandur

L230-231 In Table 1, the measured and modelled dust deposition during the years 2012 and 2013 for stations on Brúarjökull, our main area of investigation, were reported. →

[Figure]

Table 1 gives the measured and modelled dust deposition during the years 2012 and 2013 for stations on Brúarjökull, our main area of investigation.

L338 "magnitude" should probably be "order of magnitude"

L350 which seems to be overestimated → which seems to be an overestimate in the light of results presented here.

L370 supper site → upper site

L390 and L606 Grímsvötn eruption → Gjálp eruption

---

## Author Comment (AC1) · 28 Feb 2017

Dear Anonymous Referee #2,
Thank you for your letter. The comments of the associated editor and of the reviewer were most helpful and we would like to express our gratitude for the constructive support. We followed all suggestions for changes and hope to have answered all questions. Reviewer comments are listed as italic, and the following answers are below.
Best regards,
Monika Wittmann

**Changes:**
*Comment Reviewer:*

- *Figure 2 and associated discussion on page 52: Neither the figure caption nor the text is clear on exactly which period is being displayed here for the modelling results: The glaciological year 2012-2013, the entire calendar year 2013 or the part of 2013 leading up to the sampling expedition in October 2013 (data from samples collected during that expedition are displayed along with the model results in the figure). It is probably the period JD 130-283, but this should be explicitly mentioned.*

Answer: Line 120 states: We used dust measurements in snow for the year 2013, since no measurements were available for 2012, and compared them for the same time period (until October 2013) with the simulated spacial dust distribution over Vatnajökull by FLEXPART.
Since this doesn't seem to be clear enough the exact period for the flexpart model run 2013 is January 1st until October 7th, the day when the surface dust has been taken on the glacier (=DOY 280). This has been clarified in the text now and in the Figure 2 caption.

- *The two case studies on Dust events 1 and 2 (Figures 3-5) are well described and the authors present good reasons for focusing on those events, comparing measured albedo drops with modelled dust deposition. Since both are, however, spring events, it is a bit surprising that other events do not receive comparable scrutiny, like for example the summer event E5 during JD 220-227 (Fig. 3), or the September events after JD240.*

We have chosen to only describe 1, maximum 2 dust events in detail, otherwise it would have been a too long description. Therefore Table 2 and 3 are there to show all dust events and their most important parameters. Event 5 has not been chosen to be described in detail since the dust peak in B13 occurs at the same time as the highest temperature of the year (almost 5°C), and event 6 has been an exceptionally long event of 2 weeks, with a lot of precipitation and as well positive temperatures. The two events with the highest certainty that albedo drop is mainly/only caused by dust has been chosen, which cannot be guaranteed for E5 and E6.

- *L42-43 "The snow-albedo feedback, where radiation absorption is enhanced due to impurity content in snow and ice is indicated by complex processes. . ." Further clarification needed here, what is meant by "complex processes" ?*

Has been changed to: Due to impurities in snow, the albedo of the snow can be reduced. This involves direct albedo reduction by the impurities but also changes in the snow grain size triggered by the impurities especially at temperatures close to the melting point, which can strongly enhance the albedo reduction.

- *L64-66 "Iceland is one of the most active aeolian places on Earth, even though it is not situated in an arid climate (Arnalds et al., 2016). Due to the large area of sandur plains and strong winds resulting in numerous dust events." "aeolian place" is not well put, and second sentence is subordinate, meaning that it shouldn0t stand on its own.*

The sentence has been rephrased.

- *L212 Dynjgusandur → Dyngjusandur*

This has been changed

- *L230-231 In Table 1, the measured and modelled dust deposition during the years 2012 and 2013 for stations on Brúarjökull, our main area of investigation, were reported. → Table 1 gives the measured and modelled dust deposition during the years 2012 and 2013 for stations on Brúarjökull, our main area of investigation.*

This has been changed.

- *L338 "magnitude" should probably be "order of magnitude"*

This has been changed.

- *L350 which seems to be overestimated→ which seems to be an overestimate in the light of results presented here.*

This has been changed.

- *L370 supper site → upper site*

This has been changed.

- *L390 and L606 Grímsvötn eruption → Gjálp eruption*

This has been changed.

---

## Author Comment (AC2) · 2 Mar 2017

Dear Anonymous Referee #1,

Thank you for your letter. The comments of the associated editor and of the reviewer were most helpful and we would like to express our gratitude for the constructive support. We considered all suggestions for changes and hope to have answered all questions. Reviewer comments are listed as italic, and the following answers are below.

Best regards,
Monika Wittmann

General comments:

- *The comparison between observed and simulated dust depositions showed quite big differences. For example, in Figure 2 I cannot see the "similar pattern" (line 208-209) the authors refer to. Furthermore, any direct correlation between observed and simulated data is not provided. I suggest to expand the discussion about this comparison, highlighting possible causes of the differences (e.g. the timing of the field campaigns).*

We considered it best to show measurements superimposed as coloured circles on top of the FLEXPART results shown as background using the same colour scheme. This reveals also similarities in general dust deposition patterns, which would be hidden in point comparisons, such as shown in a correlation plot. The following figure shows the interpolated surface dust measurements from Dragosics et al. (2016), where it can perhaps be seen more clearly that most dust is deposited in the southwestern part of the ice cap followed by Brúarjökull, very similar to the patterns seen in Figure 2 from the FLEXPART simulations. Furthermore, in L215-229 we presented a statistical comparison and also presented possible explanations for the differences (e.g., model resolution). We considered adding a correlation (as inserted below) plot but think this would not provide any extra information.

[Figure]

The field campaigns took place at the end of the ablation season (October) and start of winter, a time at which the main dust sources are snow covered again. Thus, not much dust is expected to be mobilized after that. Also, notice that the modelling period also ends at the same time when the measurements were taken (October 7th), thus facilitating a direct comparison.

[Figure]

**Figure 1: Scatterplot of observed vs. simulated values for the stations. The Flexpart model value from the grid cell was taken where a station is located and ploted against the measurements on x axis. A regression curve is shown. The correlation coefficient is r= 0.35822342.**

- *The choice of the year 2012 should be strengthened. I don't understand why the authors did not use 2013 as a test year, since they have field data for that year. If I understand well, they are comparing field observation from 2013 with simulations of 2012. In my opinion, this needs a stronger justification. Here they are assuming that the spatial patterns of dust deposition on the ice cap are equivalent from year to year.*

Dust events modelled by FLEXPART have been compared with parameters of AWS such as albedo for 2012, and simulations of 2013 were compared to field observations of the same year (since no field observations for 2012 were available). The arguments for choosing the year 2012 have been strengthened in the text: The year 2012 was chosen for analysis because it was characterized by a negative mass balance due to warm temperatures and exceptionally low glacier albedo on Brúarjökull with a significant frequency of northerly winds. Furthermore, 2012 was not directly influenced by dust deposition from volcanic eruptions, and albedo data from weather stations were available. Dust events modelled by FLEXPART were more distinct in 2012 and agreed better with the albedo observations than in 2013. We used dust measurements in snow for the year 2013, since no measurements were available for 2012, and compared them for the same time period (until October 7 2013, DOY 280) with the simulated spatial dust distribution over Vatnajökull by FLEXPART modelled from January 1st until October 7th 2013.

- *I suggest to delete the part on MODIS data, since they are not used quantitatively in the study. In Figure 6, I cannot recognize any dust plume or deposited dust on the ice cap. If anything, other satellite (e.g. Landsat) and model products can be used to represent dust plumes and/or depositions. If you want to keep MODIS data, I suggest to compare data from the AWS with MODIS snow albedo time series (MOD10A1, MYD10A1), which could be very interesting from a remote sensing perspective.*

We admit that Figure 6 only provides qualitative information but we still think that it is helpful for the discussion of event 2. The figure has been adapted to hopefully show the dust plume clearer now. A comparison between different MODIS images shows the presence of the dust plume very clearly during event 2 over the glacier, as the brownish hues are normally not present there.

- *I suggest to add in the introduction a description of the "state" of Vatnajökull ice cap (e.g. mass balance data), in order to ensure a broader impact of the paper. Why is it important to study the impact of LAI on Vatnajokull? Is there any missing link between the temperature increase and ice melting? In the introduction, reference to the impact of LAI on Greenland (e.g. Tedesco et al. 2016 TC, Dumont et al. 2014 Nat. Geo.) could also be helpful to describe the process on ice sheets, which is more interesting for climate analysis.*

Information about Vatnajökull has been added to the manuscript introduction.

The impact of LAI with recommended references has been added to the introduction as well.

- *The Appendix is more suitable as Supplementary Information.*

The appendix has been separated from the paper as supplement.

- *The dust mobilization scheme (FLEXDUST) is based on a paper that was submitted to JGR (Groot Zwaaftink et al.). I suggest to expand the description of this scheme, sinceat the moment the reader cannot have details on it.*

Meanwhile the paper has been published:

Groot Zwaaftink, C. D., H. Grythe, H. Skov, and A. Stohl (2016), Substantial contribution of northern high-latitude sources to mineral dust in the Arctic, J. Geophys. Res. Atmos., 121, doi:10.1002/2016JD025482.

- *I'm not an expert in mass balance modeling, so I don't have specific comments on it. In any case, the impact on mass balance is evaluated using only two AWS dataset. Is it possible to extrapolate this information to the whole Vatnajokull ice cap? or the impact of mineral dust could be limited to some areas of the glacier?*

    In case of dust on the surface the effect would be similar in other areas of Vatnajökull. The two sites investigated are close to equilibrium line altitude and high in the accumulation zone; they thus represent areas where snow is melted. Dust in the ablation zone in early spring would have a similar effect, increasing the melt rate of the snow cover and thus also exposing the dirty ice surface earlier, so in addition increase the total ice melt. If however the dust precipitated on ice in the ablation zone the effect will be less since the dust washes off quickly.

    Some parts of Vatnajökull are more prone to dust storms; NE Vatnajökull probably has the highest likelihood for this. But more research is needed into that topic.

Specific comments:

- *Figure 1: It is very small and labels are difficult to read. I suggest to enlarge Fig.1A and to remove Fig.1B since the details are not relevant for this study. As far as I understand, only data from B13 and B16 (+ the firn core) are used in this work.*

Changes have been made as suggested.

- *Figure 2: I suggest to use a linear scale bar palette for dust deposition.*
- *line 56 Here briefly describe the impact of LAI on Greenland Ice sheets.*
- *line 76: delete "want".*
- *line 77: please rephrase this sentence.*
- *line 92: "Snow cover inhibits the dust emission": explain better or remove this sentence*

Changes have been made as suggested.

- *line 111: Why did you choose this treshold? Please explain.*

A minimum modelled concentration of 6 μg m-3 over at least two days was defined as dust event because this showed the best fit with correlation of albedo drops presented in Fig. 3. With this threshold, events (in Table 3) could be selected where an albedo change could be expected. Lower concentrations or such with a shorter duration do not seem to have a significant impact on albedo. The following figure also shows how well modelled concentration and depositions fit together:

[Figure]

- *line 116-117: here some references are needed. 2012 was an extreme year in the northern hemisphere, with strong melting in Greenland (e.g. Nghiem et al. 2012 GRL).*
- *line 118-121: here I don't understand why you chose 2012 as a test year.*

As mentioned above, this paragraph has been rewritten.

- *line 120: replace "spacial" with "spatial".*

Changes have been made as suggested.

- *line 128-130: remove this part on MODIS data.*

Changes have been made as suggested.

- *line 154: "Albedo is a key variable in the surface energy balance and used to calculate melting", change with : "Albedo is a key variable in the surface energy balance and it is used to calculate ice melting"*
- *line 160: delete "may then be" and replace with "it is"*
- *line 163-165: please rephrase this sentence*

Changes have been made as suggested.

- *line 173: I personally don't like reference to paper in preparation.*

Schmidt et al. has been submitted now to The Cryosphere!

- *line 181: delete "seemed" and replace with "is assumed to be"*

Changes have been made as suggested.

- *line 181- 183: did you remove these data? please explain*

Measured albedo values above 1 are set to 1. As mentioned in the paper this can be explained due to the high solar zenith angle, multiple reflections in autumn, and instrumental error.

- *line 188: delete "seemed" and replace with "was"*
- *line 188-190 Unsupported statements. Add some reference or lose the phrase*
- *line 194: delete "(not ice)"*

Changes have been made as suggested.

- *line 194: add more details on this point, what do you mean with "extreme year"?*

The sentence has been changed.

- *line 194-195: this is a fundamental aspect. Decoupling the effect of dust and other meteoclimatic forcing is a big issue. Explain better possible influence on your estimations*

Has been changed to: Since 2012 was a year of very warm temperatures and negative mass balance, not only deposition during dust events influenced the albedo and energy balance. Warm and dry periods with northerly winds increased the possibility for dust events to occur. Due to the negative mass balance the exposed darker firn layer lowered the albedo additionally to surface dust.

- *line 198-205: these are results, integrate in Section 3.*

Has been changed to section 3.

- *line 208-209: I can't see this "similar pattern". It looks like observational points don't show a marked spatial correlation, which is present in model simulations. This is supported by the IDW map showed in Figure 6 of Dragosics et al. 2016 (Arab. J. Geosci.), which shows "bulls eye" that are typical errors of spatial interpolation of uncorrelated data. I would expect that on a large ice cap such as the Vatnajökull, dust concentration on surface snow should in theory feature some spatial structure. Probably the impact of snow falls, melting and run off can redistribute the dust concentration. Only 16 samples on a large area (8000 km2) are too few to capture these complex processes of deposition and redistribution. Please discuss these aspects in the paper.*

The 'similar pattern' was already explained above. Meanwhile, comparisons between measurecompared and modelled dust deposition have been compared and all show this pattern with most dust in the SW followed by the north (this will be published in a separate paper). Of course 16 sample locations is a limited amount compared to the size of the ice cap, but fieldwork possibilities and finances are limited. As I will present in my next paper which is in preparation for the journal Jökull, firn cores have shown that dust deposition can have local effects and therefore has large uncertainties because dust can be washed away by rain or melt in the ablation area, can get redistributed by wind or mixed with snow. Also visits to the ice cap have shown that dust often is not deposited evenly on the surface and appears rather patchy. Since this paper has its main focus on the impact on albedo and following energy balance, and the resolution of FLEXPART cannot catch such local effects, these effects are part of a different paper and won't be explained here further.

- *line 226: Confused sentence. Please rephrase*

The sentence was rephrased

- *line 232: delete "however, the order of magnitude was captured correctly", you already said that in line 229*

The sentence was changed.

- *line 259: I don't see the dust cloud, nor the dust deposition from Figure 6.*

As mentioned above, the figure has been adapted, so hopefully the dust plume is better visible now.

- *line 284: use "from .. to .." in both parethesis*

This was changed

- *line 306: replace "precipitation" with "deposition"*

*This was changed*

- *line 324: confused sentence, please explain.*

*This was changed*

- *line 328-331: This should be better explained. BC cannot be excluded since its impact on snow is hardly visible with naked eyes (Warren 2013 JGR).*

References have been added to this statement. Recent unpublished experiments by Outi Meinander on BC in Icelandic dust has shown that since in Iceland, there are not many BC sources, and snow and ice are not expected to contain high concentrations of BC, her first results confirm this assumption. She will present at the European Geoscience Union General Assembly this year her findings of high concentrations of organic carbon in Iceland 2016 dust samples.

- *line 370: delete the "s" from "supper"*

This was changed

---

## Author Comment (AC3) · 2 Mar 2017

[revised manuscript text omitted]


---

## Author Comment (AC4) · 2 Mar 2017

The comment was uploaded in the form of a supplement:
http://www.the-cryosphere-discuss.net/tc-2016-205/tc-2016-205-AC4-supplement.pdf

---

## Author Comment (AC5) · 2 Mar 2017

**Supplementary Information**

[revised manuscript text omitted]

---

## Author Comment (AC6) · 2 Mar 2017

The comment was uploaded in the form of a supplement:
http://www.the-cryosphere-discuss.net/tc-2016-205/tc-2016-205-AC6-supplement.pdf